# New Cretaceous Bugs from Northeastern China Imply the Systematic Position of Pachymeridiidae (Hemiptera: Heteroptera) [note 1]

**DOI:** 10.3390/insects13080689

**Published:** 2022-07-31

**Authors:** Rui Dai, Sile Du, Dong Ren, Yunzhi Yao

**Affiliations:** 1College of Life Sciences, Capital Normal University, Beijing 100048, China; dairui19971030@163.com (R.D.); rendong@cnu.edu.cn (D.R.); 2Science and Technology Research Center of China Customs, Animal Quarantine Intitute, Beijing 101101, China; sile0228@163.com

**Keywords:** Yixian Formation, Pentatomomorpha, Mesozoic, costal fracture, pulvillus

## Abstract

**Simple Summary:**

A new genus and species of Pachymeridiidae is described from the Early Cretaceous of Northeastern China. The structure of the well-developed coastal facture, claws, and male parameres are preserved. The documentation of these structures is provided for exploring the systematic position of Pachymeridiidae and the evolution of the costal fracture in Heteroptera.

**Abstract:**

*Varicapitatus sinuolatus* gen. et sp. nov. is described from the Yixian Formation of Northeastern China. Based on the new specimens, this study discusses the morphological characteristics and taxonomic position of Pachymeridiidae: Pachymeridiidae belongs to Pentatomomorpha and is more closely related to Lygaeoidea. In Heteroptera, the costal fracture of the forewing is a homoplastic characteristic, already evolved independently several times among most taxa before the Early Cretaceous. The pulvillus under the claw in Pentatomomorpha is also a homoplastic characteristic. In Pachymeridiidae, the forewing costal fracture and absence of pulvillus can be recognized as an independent evolution or convergence, implying that pachymeridiids may have different flight and crawling behaviors distinct from other Pentatomomorpha groups.

## 1. Introduction

Pachymeridiidae, an extinct family, comprising 28 genera and 43 species, has been recorded in Russia, China, Germany, England, and Kazakhstan. The geologic period is throughout the Mesozoic, from the Triassic to the Cretaceous [1,2,3,4,5,6,7,8,9,10,11,12,13,14,15,16]. Geinitz [2] described *Pachymeridium dubium* Geniitz 1880 under Lygaeoidea but did not elevate it to family. Handlirsch [3] established Pachymeridiidae according to *Pachymeridium dubium* Geniitz 1880 and designated it as the type species. Nevertheless, Handlirsch did not mention the attribution of Pachymeridiidae. Until 1977, Popov and Wootton [7] placed this family under the infraorder Pentatomomorpha, based on the venation of the forewing similar to Idiostolidae. However, the forewing with the costal fracture and absence of pulvillus under the claw make Pachymeridiidae significantly distinct from extant Pentatomomorpha. Thus, in recent years, some scholars have continued to dispute the phylogenetic position of Pachymeridiidae [17,18,19]. Especially, Schuh and Weirauch [19] proposed that Pachymeridiidae is tentatively assigned to Trichophora, but key distinguishing characteristics supporting its placement in this clade or even in Pentatomomorpha are still lacking.

Most of the genera in Pachymeridiidae were established in the last century. Due to limited research conditions or poor preservation of specimens, we cannot obtain more detailed information from species reported before, which makes research on the phylogenetic position of Pachymeridiidae difficult. Fortunately, we have collected some well-preserved fossils specimens from the Yixian Formation in recent years, which not only demonstrate higher species diversity in Pachymeridiidae, but also maintained more detailed characteristics that provide additional evidence to analyze the phylogenetic position of Pachymeridiidae. Herein, we have established a new genus and species based on the well-developed coastal facture, claws, and male parameres. In addition, we discuss the key characteristics and the taxonomic position of Pachymeridiidae.

## 2. Materials and Methods

The seven specimens used in this paper were collected from the Yixian Formation (includes two male specimens and five female specimens), which are located in Huangbanjigou Village in Beipiao City, Liaoning Province, China. The geological age is about 125 Ma, and considered to be Lower Cretaceous [20,21]. This formation contains numerous, diverse insect fossils including Hymenoptera, Hemiptera, Coleoptera, Siphonaptera, etc. [22,23,24,25,26,27].

The new material was stored at the Key Laboratory of Insect Evolution and Environmental Changes, College of Life Sciences, Capital Normal University, Beijing, China (CNUB, Curator: Yunzhi Yao). the specimens were observed, photographed, and line drawn under a Nikon SMZ-25 microscope with a Nikon DS-Ri 2 digital camera system. The classification system and morphological terminology used are based on Schuh and Weirauch [19]. All measurements are in millimeters.

## 3. Results

Systematic palaeontology.

Order Hemiptera Linnaeus, 1758.

Suborder Heteroptera Latreille, 1810.

Infraorder Pentatomomorpha Leston, Pendergrast and Southwood, 1954.

Family Pachymeridiidae Handlirsch, 1906.

Genus *Varicapitatus* Dai, Du, and Yao, gen. nov.

urn:lsid:zoobank.org:act:5F6C7560-A9B1-4CAD-AD0D-93254AE11755

Etymology. The generic name is a combination of the Latin words ‘*vari*-’ (strange) and ‘*capitatus*’ (head), in reference to the unique head shape. The gender is masculine.

Type species. *Varicapitatus sinuolatus* gen. et sp. nov. (Figure 1 and Figure 2).

Diagnosis. Body small and elongated (about 4 mm). Head square, truncate (Figure 1). Bucculae small, not reaching anterior ocular margins (Figure 2A). Antennal segment I barrel shaped and exceeding apex of head (Figure 2F). Labium reaching hind coxae. Preocular tubercle present (Figure 2A). Pronotum punctured, collar present. Corium costal margin membranous and posterior margin punctured; costal fracture connected to medial fracture (Figure 2E). Clavus entirely covered punctures, claval commissure longer than the scutellum. Scutellum small, not surpassing one-half the width of pronotum, lateral margins bulge. Pulvillus absent (Figure 2D). Connexivum on segments III–VII (Figure 1C,F). Ovipositor long, gonoplac present (Figure 2B). Male genitalia symmetrical (Figure 2C).

Remarks. *Varicapitatus* gen. nov. may be attributed to Pachymeridiidae by the presence of several characters: rostrum 4-segmented, slender, segment I visible, corium with deep costal fracture, ovipositor long, extending through the last three abdominal segments. Furthermore, the following characteristics of *Varicapitatus* gen. nov. allow it to be clearly distinguished from other genera in Pachymeridiidae: head transverse, antennal segment I barrel shaped, costal fracture connected to medial fracture, scutellum small, lateral margins bulge.

*Varicapitatus* gen. nov., *Beipiaocoris* Yao, Cai and Ren, 2008 and *Bellicoris* Yao, Cai and Ren, 2008; are both from the Yixian Formation and the former is distinctly distinguishable from the latter two by its small body size of about 4 mm (vs. two body sizes of about 8 mm), as well as by its square-shaped head (vs. pentagonal head), preocular tubercle present (vs. no preocular tubercle), labium reaching hind coxae (vs. rostrum only reaching to middle coxae), pronotum collar present (vs. pronotum collar absent).

*Positocoris* Popov, 1990, *Pronotaphanus* Popov, 1990 and *Takshania* Popov, 1990, all from the Karatau Formation of Siberia, Russia, are also small types (no more than 5 mm) in Pachymeridiidae. However, it is easily possible to distinguish the new genera from them based on the following characters: head transverse, anterior margin straight (vs. head pentagonal), antennal segment I thickened beyond anterior end of head (vs. antennal segment I not exceeding anterior end of head), scutellum small, not surpassing one-half the width of pronotum. (vs. large scutellum, apparently exceeds 2/3 of width of pronotum.)

*Varicapitatus* gen. nov. has a distinctive medial fracture that joins the costal fracture in a complete arc, whereas in Pachymeridiidae, only the oldest, *Pachymerus* Giebel, 1856, found in the Late Triassic, is mentioned as having the medial fracture, whereas not in subsequent genera. However, there is a clear distinction between the two, as in *Pachymerus* Giebel, 1856, although the medial fracture is obvious, there is no obvious costal fracture, and the head is pentagonal, whereas the head of the new genus is square, making the two easily distinguishable.


***Varicapitatus sinuolatus* sp. nov.**


urn:lsid:zoobank.org:act:DDBCC10E-9C2F-4A5B-8CB0-911278AF983B

Etymology. Species name is derived from the Latin word ‘*sinuolatus*’ (finely curved), referring to the species with finely curved male parameres. The gender is masculine.

Type material. Holotype, male: CNU-HET-LB2022003. Paratype, female: CNU-HET-LB2022004–9. 

Locality and Horizon. Huangbanjigou, Chaomidian Village, Beipiao City, Liaoning Province, China (N 41°18.979′, E 119°14.318′), Yixian Formation, Lower Cretaceous.

Diagnosis. Segment I of antenna shortest and thickest, segment II longest, segment IV fusiform (Figure 2F). Eyes round, clearly convex at the lateral margin of the head, and distinctly removed from anterior pronotal margin. Labium segment III longest, segment I shortest, and segment II as long as segment IV. Pronotum trapezoidal, anterior distinctly narrowed, posterior angles feebly rounded. Segment II of tarsus shortest, segment I as long as III. Paramere styles slender, no teeth, no raised, somewhat narrowed and curved from middle to apex (Figure 2B).

Description. Body punctured, about 2.7 times (male) or 2.1 times (female) as long as wide (3.8–4.3 mm).

Head: Short head, transverse, 2.3 times as wide as long. Antennae inserted anterior margin of eye, longer than head and pronotum combined, segment III subequal to IV, segment II 2.1 times as long as I, about 1.2 times as long as III and IV. Ocelli situated at front level of posterior margins of eyes, interocellar space narrower than interocular space. Labium slender, segment II 1.2 times as long as I, segment III 1.7 times as long as II and IV.

Thorax: Pronotum longer than head, about twice as long as head, moderately transverse, nearly 1.7 times as wide as long, anterior margin narrow, posterior margin wide, posterior 2.2 times as long as anterior. Scutellum small, about 1/10th of the body length, transverse, and about 1.7 times as wide as long. Mesosternum subequal to metasternum in length, metasternum with convex posterior edge. 

Legs: Coxae rounded, trochanter rounded triangle, all femora thickened, about 2–3 times as thick as corresponding tibiae, fore femur as long as tibiae, middle tibiae slightly longer than femur, hind legs distinctly longer than fore and mid-legs, hind tibiae longer than femur, I and III tarsal segments about 2.3 times as long as segment II. The forewing is about 1.4 times as long as the anterior margin of the corium, costal fracture joined to middle fracture at corium 2/3. Clavus wide and large, about 0.4 times as long as the forewing, and 4.4 times longer than wide, Claval commissure approximately 1.2 times as long as scutellum.

Abdomen: Abdomen oval, connexivum narrowed. Sternum VII of female widest and split by ovipositor. Ovipositor long, one-third as long as body.

Dimensions (mm; holotype data in brackets). Body length 3.83–4.33 (3.83); maximal width of body 1.57–1.79 (1.79); head length 0.41–0.48 (0.41), width 0.98–1.01 (0.98); length antennal segments I–IV: 0.23–0.28, 0.53–0.71, 0.43–0.61, 0.47–0.58 (0.26, 0.56, 0.46, 0.47); length rostral segments I–IV:0.39, 0.45, 0.75, 0.43; length pronotum 0.91–0.93 (0.92), width 1.47–1.62 (1.62); length scutellum 0.38–0.44 (0.44), width 0.69–0.75 (0.75); length hemelytron 2.84–2.96 (2.84), length anterior margin of corium 2.01–2.32 (2.01), length clavus 1.16–1.29 (1.29),width 0.28–0.29 (0.29); length fore leg: femur 0.74, width 0.22, tibia 0.79, width 0.07, tarsomeres I–III: 0.16, 0.07, 0.18; length middle leg: femur 0.81, width 0.18, tibia 0.91, width 0.08, tarsomeres I–III: 0.17, 0.07, 0.17; length hind leg: femur 1.17, width 0.23, tibia 1.34, width 0.08, tarsomeres I–III: 0.21, 0.11, 0.19.

## 4. Discussion

The taxonomic position of the Pachymeridiidae is an issue of intensive debate. Popov and Wootton [7] proposed that Pachymeridiidae is more closely related to Coreoidea (*sensu lato*) and may be ancestral to them. Belayeva et al. [17] came to the same view on account of Pachymeridiidae retaining the costal fracture. Yao et al. [18] utilized the morphological data for the first time to analysis the phylogeny relationship of Pentatomomorpha, and showed that Pachymeridiidae and Idiostolidae are sister groups, and further argued that Pachymeridiidae is related to Coreoidea (*sensu lato*). At present, the monophyly of Coreoidea (*sensu lato*), which consist by Lygaeoidea, Coreoidea, Pyrrhocoridea, and Idiostoloidea, is supported by many scholars [28,29,30,31]. With combined fossil records and new specimens, we agree the Pachymeridiidae is more closely related to Coreoidea (*sensu lato*) for several reasons. At first, pachymeridiids have pentagonal heads, the anteclypeus more developed than the mandibular plate [12,14,16] and are similar to the common type of Coreoidea (*sensu lato*) [19]. Although the head shape of the new genus is atypical, the square-headed, truncate types is still similar to Malcidae (Lygaeoidea) [32]. Besides, veins on the corium of Pachymeridiidae are very similar to that of *Idiostolus insularis* Berg, 1883 (Idiostoloidea) [7,33,34], and the membrane with some free veins is also typical of coreoids vein [7]. Despite that the veins of *Varicapitatus* gen. nov. may not be visible for preservation reasons, part of the membranous corium with punctate still resembles Ninidae (Lygaeoidea) [35,36]. In addition, the new genus and *Bellicoris* Yao, Cai and Ren, 2008 alike own a complete seventh-connexival, the characters also apply to most of Coreoidea (*sensu lato*), except for Idiostoloidea [15,33,37]. More importantly, we can observe the morphology of the male paramere in our specimens for the first time, simple, with no teeth, not raised, somewhat narrowed and curved from middle to apex, which is similar to Lygaeoidea, especially *Mizaldus* Krüger, 2019 (Figure 3) [38,39]. Females have laciniate ovipositors, which extend through the last three abdominal segments. The morphological structure of ovipositors is also similar to extant Lygaeoidea [13,40,41]. Above all, we consider that the phenotype traits of Pachymeridiidae are similar to Coreoidea (*sensu lato*), especially to Lygaeoidea, to which it might be closely related, thus it should be assigned to Pentatomomorpha. As for the forewing with costal fracture and absence of pulvillus under the claw, these traits make Pachymeridiidae different from other extant Pentatomomorpha, but we do not consider that this affects its taxonomic position for the following reasons:

According to the research on functional wing morphology, the costal fracture of true bugs may be acting as a shock absorber when encountering collisions or wings deformation in flight [42,43]. At the same time, it plays an auxiliary role in wing folding [44,45,46]. Shcherbakov [47] also proposed that the extension of the costal fracture forms a thickened cuneus in some taxa, which can have a function similar to pterostigma. Based on morphological and molecular data of the phylogenetic analysis of the extant Heteroptera, the result is the costal fracture discretely distributed in some taxa of Dipsocoromorpha, Enicocephalomorpha, Leptopodomorpha, Nepomorpha (only the superfamily Ochteroidea), and Cimicomorpha [30,48]. Therefore, the costal fracture is an important functional characteristic closely related to flight, evolving independently at least five times in Heteroptera (Figure 4).

The known phenotype of the costal fracture may be divided into two types: first, an arched, strongly inclined, ending not far from the apex of medial fracture; second, a transverse, perpendicular to the anterior margin of the corium, partially extending forward to form cuneus [47]. Throughout the appearance, the time of the costal fracture in Heteroptera and the divergence time of two phenotypes, we find: the earliest known costal fracture was found in *Arlecoris louisi* Shcherbakov, 2010 (Naucoroidea) from the Middle Triassic, with a more oblique arch shape (Figure 5A) [49]. All species in Corixidae and Belostomatidae from the Late Triassic also have an arch shaped costal fracture (Figure 5B,C), while the costal fracture is absent in these extant taxa [13]. The priod after the Jurassic, such as Archegocimicidae, Saldidae, and Ochteroidae, also showed this arched costal fracture (Figure 5D–G) [50,51]. Until the Early Cretaceous, transverse types appeared in flower bugs and naboids (Figure 5H,I) [47,52]. However, several genera in flower bugs from the late Mesozoic costal fracture were absent, such as *Crassicerus* Tang, Yao and Ren, 2015, *Longilanceolatus* Tang, Yao and Ren, 2015. Nevertheless, most taxa of flower bugs have the significant transverse costal fracture at present [53]. More interestingly, the extant ochteroids have well-developed costal fracture, whereas, in the Early Cretaceous, not all species have this characteristic [50]. The available fossils suggest that the costal fracture in Heteroptera is likely to be a rapidly evolving, unstable characteristic and that its occurrence in various groups shows great complexity. Therefore, the costal fracture has certain limitations in the classification of fossil Heteroptera, and it should be carefully selected as a distinguishing characteristic. Two phenotypes of the costal fracture diverged in the Early Cretaceous and had already evolved independently several times between most taxa of Heteroptera during this period and even earlier. All published species of Pachymeridiidae have an ‘arched’ costal fracture (Figure 5E), suggesting that this is a stable form in the forewing and could serve as a diagnostic characteristic for the family.

All of the extant Pentatomomorpha have pulvillus, which is considered not only as evidence in favor of its monophyly [30,54], but also as a characteristic to distinguish Pentatomomorpha and Cimicomorpha. However, some groups in Cimicomorpha, such as Miridae and Tingidae, also have pulvillus [55]. At the same time, the relationship between Pentatomomorpha and Cimicomorpha is still controversial, with some scholars believing that they are sister groups [31,56,57,58,59,60,61], while others believe that they form a paraphyletic group [62,63,64,65]. Actually, regardless of their relationship, the pulvillus has multiple origins in Pentatomomorpha and Cimicomorpha. Thus, pulvillus is not as synapomorphy characteristic in Pentatomomorpha, although most species have it. In all fossil records—and we have re-examined nearly 1000 Mesozoic specimens of Pachymeridiidae from China—it is noted that pulvillus is absent in Pachymeridiidae. However, in both of the fossil species of Hemiptera and Mecoptera in the same locality, we find the pulvillus and even the hairs are well preserved [53,66,67]. Apart from reasons for burial, the absence of the pulvillus is an important taxonomic feature to distinguish Pachymeridiidae from other groups of Pentatomomorpha.

## 5. Conclusions

*Varicapitatus sinuolatus* gen. et sp. nov is assigned to Pachymeridiidae, which belong to Pentatomomorpha and may be related to Lygaeoidea. In Heteroptera, the costal fracture of the forewing as an important functional characteristic closely related to flight, already evolved independently several times among most taxa before the Early Cretaceous. It is a variable characteristic that evolves rapidly, so it should be chosen with care when classifying Heteroptera fossil taxa. However, the costal fracture in Pachymeridiidae is a stable characteristic and went through an independent evolution. The pulvillus under the claw is of multiple origins in Pentatomomorpha and Cimicomorpha. In Pentatomomorpha, the pulvillus is a homoplastic characteristic, which may not be evidence in support of its monophyly. The absence of pulvillus in Pachymeridiidae may also be independently evolved. It is obvious that the costal fracture of the forewing and the absence of pulvillus were characteristics important for separating Pachymeridiidae from other bugs in Pentatomomorpha, which suggest that they may have different flight and crawl behaviors distinct from other groups.

## Figures and Tables

**Figure 1 insects-13-00689-f001:**
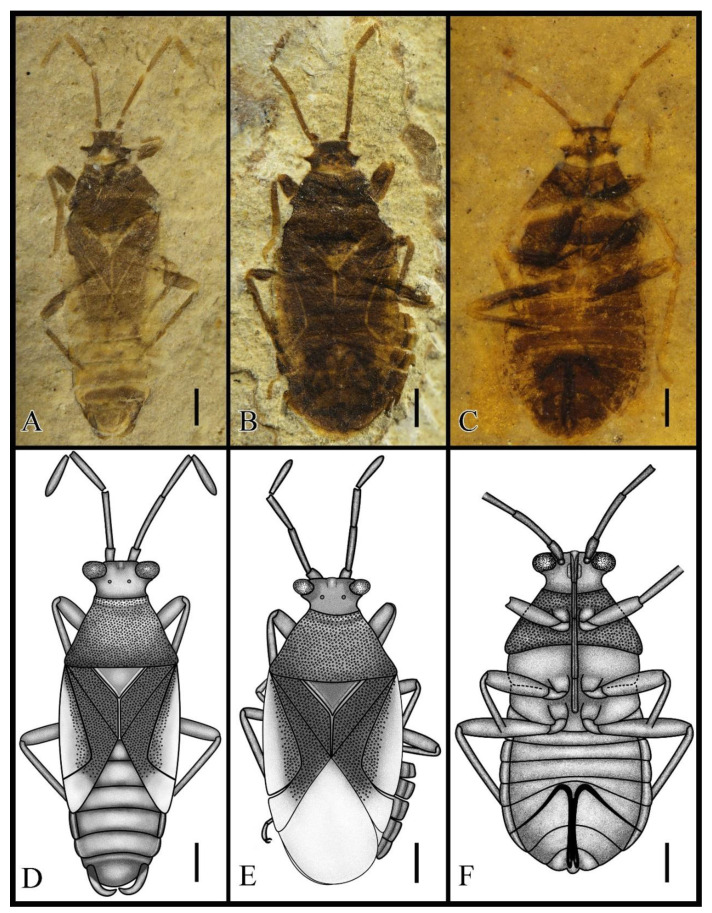
*Varicapitatus sinuolatus* gen. et sp. nov. from the Early Cretaceous of Huangbanjigou, China. (**A**,**D**) Holotype, male, CNU-HET-LB2022003. Photograph and line drawing in dorsal view. (**B**,**E**) Paratype, female, CNU-HET-LB2022004. Photograph and line drawing in dorsal view. (**C**,**F**) Paratype, female, CNU-HET-LB2022005. Photograph and line drawing in ventral view. Scale bars: (**A**–**F**) = 0.5 mm.

**Figure 2 insects-13-00689-f002:**
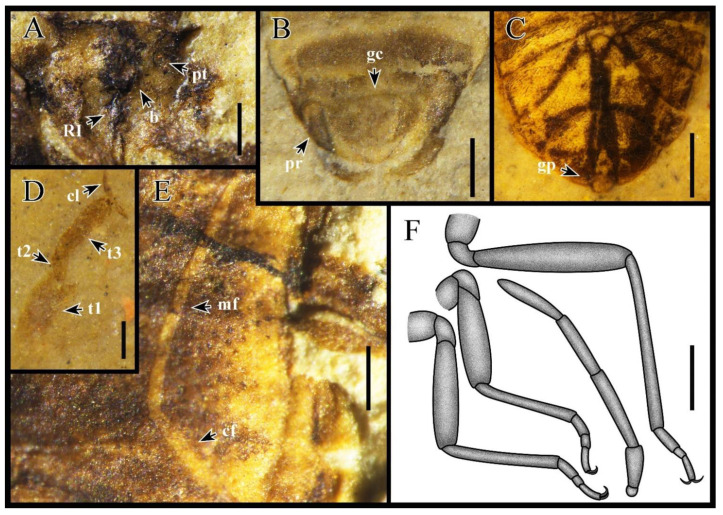
*Varicapitatus sinuolatus* gen. et sp. nov. with morphological details. (**A**) Head, paratype, female, CNU-HET-LB2022006. (**B**) Dorsal view of male genitalia, holotype, male, CNU-HET-LB2022003. (**C**) Ventral view of female genitalia, paratype, female, CNU-HET-LB2022007. (**D**) Tarsal, paratype, female, CNU-HET-LB2022008. (**E**) Forewing, paratype, female, CNU-HET-LB2022004. (**F**) Line drawing of legs and antennae female, CNU-HET-LB2022009; CNU-HET-LB2022006. Abbreviations are as follows: RI, first rostral segment; b, buccula; pt, preocular tubercle; pr, paramere; gc, genital capsule; gp, gonoplac; t1–3, the first to third tarsal segment; cl, claw; mf, medial fracture; cf, costal fracture. Scale bars: (**A**) = 0.2 mm; (**B**) = 0.3 mm; (**C**,**F**) = 0.5 mm; (**D**) = 0.1 mm; (**E**) = 0.2 mm.

**Figure 3 insects-13-00689-f003:**
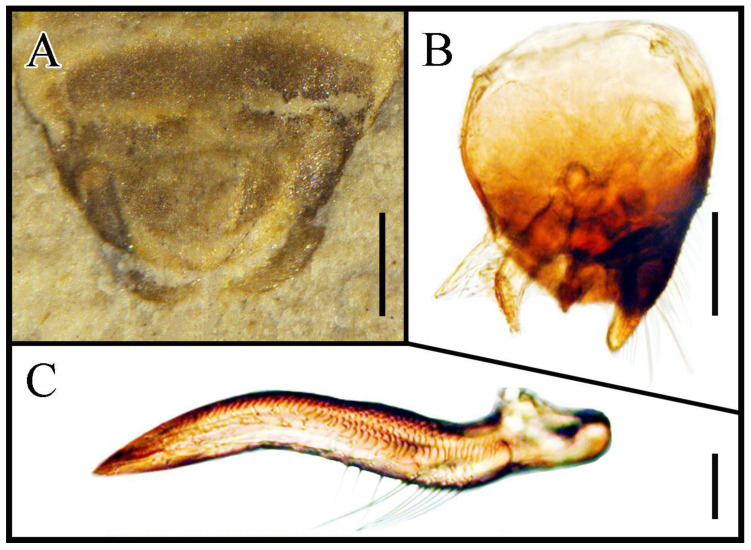
Comparison of male genitalia. (**A**) Pachymeridiidae *Varicapitatus sinuolatus* gen. et sp. nov. male genitalia, dorsal view, male, CNU-HET-LB2022003. (**B**) Lygaeoidea Rhyparochromidae *Mizaldus sylvaticus* Krüger, 2019 male genitalia, lateral view. (**C**) Lygaeoidea Rhyparochromidae *Mizaldus carvalhoi* Slater, 1995 male Paramere. (Krüger, 2019). Scale bars: (**A**) = 0.2 mm; (**B**) = 0.1 mm; (**C**) = 0.05 mm.

**Figure 4 insects-13-00689-f004:**
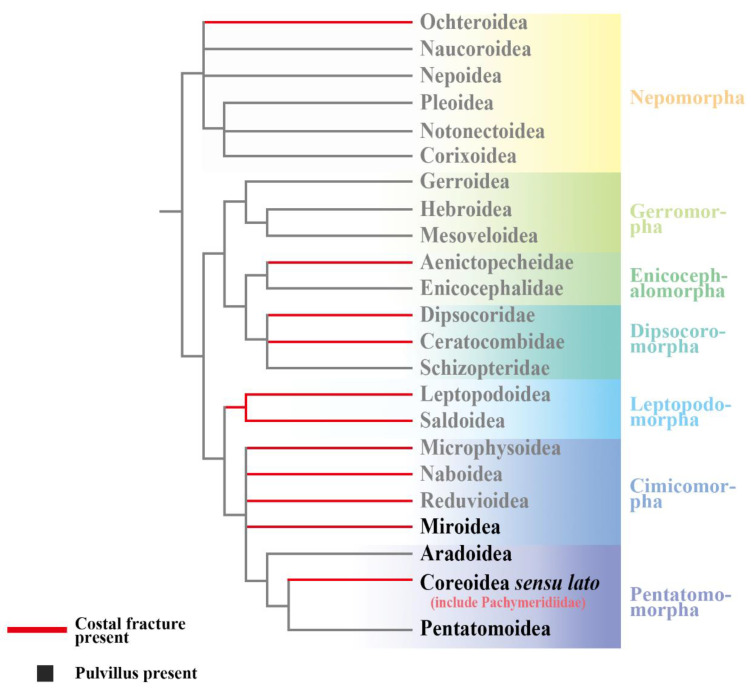
Phylogenetic hypothesis for Heteroptera (modified from [30]). Redline shows the costal fracture present, boldface shows the pulvillus present.

**Figure 5 insects-13-00689-f005:**
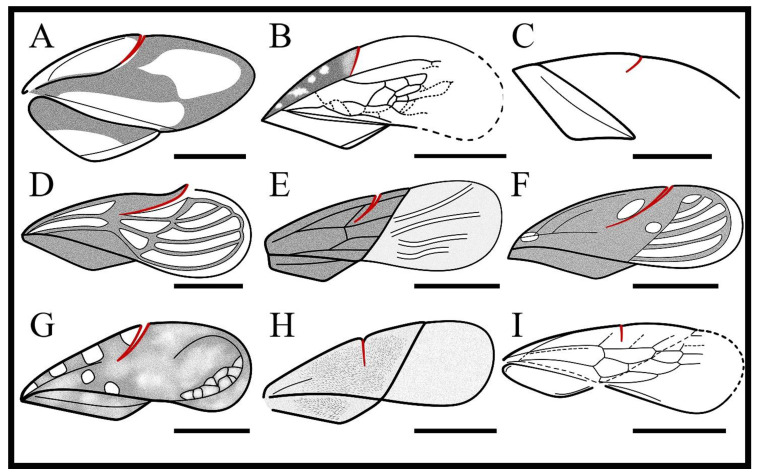
Redrawn from the forewings of other taxa of Heteroptera with changes. (**A**) Naucoroidea *Arlecoris louisi* Shcherbakov, 2010. (**B**) Belostomatidae *Tarsabedus menkei* Popov, Dolling and Whalley, 1994. (**C**) Corixidae *Liassocorixa dorsetica* Popov, Dolling and Whalley, 1994. (**D**) Archegocimicidae *Mesolygaeus laiyangensis* Ping, 1928 (Zhang et al., 2014). (**E**) Pachymeridiidae *Bellicoris mirabilis* Yao, Cai and Ren, 2008. (**F**) Saldidae *Brevrimatus pulchalifer* Zhang, Yao and Ren, 2011. (**G**) Ochteroidae *Angulochterus quatrimaculatus* Yao, Zhang and Ren, 2011. (**H**) Vetanthocoridae *Punctivetanthocoris pubens* Tang, Yao and Ren, 2016. (**I**) Nabidae *Juracipeda popovi* Shcherbakov, 2007. Scale bars: (**A**) = 1 mm; (**B**) = 10 mm; (**C**,**E**–**H**) = 2 mm; (**D**,**I**) = 1.5 mm. Redline shows the costal fracture.

## Data Availability

All data is provided in the manuscript.

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
