# Peer review of "New Cretaceous Bugs from Northeastern China Imply the Systematic Position of Pachymeridiidae (Hemiptera: Heteroptera)"

_insects, 2022, doi:10.3390/insects13080689_

Round 1

Reviewer 1 Report

This is an important contribution as it helps clarify the taxonomic position of a group of fossil true bugs, as well as contributes to better understand the morphology of this previously poorly known group; it includes fine photographs and illustrations.  Several minor editorial suggestions are marked in an attached Word file, please carefully revise once more the whole text.  Not clear to me whether only one male holotype and one female paratype are known or were studied, or measured, if possible please clarify, neither is clear wheter a formal material examined section is included (or it is not necessary), please consider this.  The mention of Mecoptera (with pulvillus, a non-related insect group) seems a bit odd to me, please reconsider or justify.

Reviewer 2 Report

Nice contribution to our understanding our the extinct bug family Pachymeridiidae and new data on character evolution that have important implication for the systeamtic placement of the extinct family. The ms can be accepted for publication after a minor revision. All my detailed comments are annotated in the pdf. 

Reviewer 3 Report

The authors described a new genus and species of Pachymeridiidae from Early Cretaceous and discussed phylogenetic relationship of related taxa. It is worthy to be published in Insects after modified. I have commented directly in the manuscript. But the authors should pay attention to following two points.

1.     The quality of three drawings in Figure 1 is not good enough and should be deleted.

2.      The reference format must strictly follow the journal’s.

Author Response

请参阅附件。
